# Tracking Separate Contributions of Diesel and Gasoline Vehicles to Roadside PM$_{2.5}$ Through Online Monitoring of Volatile Organic Compounds and PM$_{2.5}$ Organic and Elemental Carbon: A Six-Year Study in Hong Kong

5 Yee Ka Wong[1], X.H. Hilda Huang[1], Peter K.K. Louie[2], Alfred L.C. Yu[2], Damgy H.L. Chan[2], Jian Zhen Yu[1,3]

[1]Division of Environment & Sustainability, Hong Kong University of Science & Technology, Clear Water Bay, Kowloon, Hong Kong
[2]Hong Kong Environmental Protection Department, 47/F, Revenue Tower, 5 Gloucester Road, Wan Chai, Hong Kong
10 [3]Department of Chemistry, Hong Kong University of Science & Technology, Clear Water Bay, Kowloon, Hong Kong

*Correspondence to*: Jian Zhen Yu (jian.yu@ust.hk)

**Abstract.** Vehicular emissions contribute a significant portion to fine particulate matter (PM$_{2.5}$) air pollution in urban areas. Knowledge of the relative contribution of gasoline versus diesel powered vehicles is highly policy relevant and yet there lacks an effective observation-based method to determine this quantity, especially for its robust tracking over a period of 15 years. In this work, we present an approach to track separate contributions by gasoline and diesel vehicles through positive matrix factorization (PMF) analysis of online monitoring data measurable by relatively inexpensive analytical instruments. They are PM$_{2.5}$ organic and elemental carbon (OC and EC), C$_2$–C$_9$ volatile organic compounds (VOCs) (e.g., pentanes, benzene, xylenes, etc.) and nitrogen oxides concentrations. The method was applied to monitoring data spanning over six years between 2011 and 2017 in a roadside environment in Hong Kong. We found that diesel vehicles accounted for ∼ 70–90 20 % of the vehicular PM$_{2.5}$ (PM$_{vehicle}$) over the years and the remaining from gasoline vehicles. The diesel PM$_{vehicle}$ during a truck- and a bus-dominated periods showed declining trends, in coincidence with control efforts targeting at diesel commercial vehicles and franchised buses in the intervening period. The combined PM$_{vehicle}$ from diesel and gasoline vehicles by PMF agrees well with an independent estimate by the EC-tracer method, both confirming PM$_{vehicle}$ contributed significantly to the PM$_{2.5}$ in this urban environment (∼ 4–8 µg m$^{-3}$, representing 30–60 % in summer and 10–20 % in winter). 25 Our work shows that long-term monitoring of roadside VOCs and PM$_{2.5}$ OC and EC is effective for tracking gaseous and PM pollutants from different vehicle categories. This work also demonstrates the value of an evidence-based approach in support of effective control policy formulation.

## 1 Introduction

Vehicular emissions (VE) are among the major sources of air pollution in the urban environment. Major constituents in VE 30 include nitrogen oxides (NO$_x$), carbon monoxide (CO), volatile organic compounds (VOCs) and fine particulate matter (PM$_{2.5}$). Two primary components in vehicular PM$_{2.5}$ (PM$_{vehicle}$) are elemental carbon (EC) and organic matter (OM) (Kleeman et al., 2000; Chow et al., 2011). Growing evidence has shown exposure to VE affects human health (Peters et al., 2004; Beelen et al., 2008; Benbrahim-Tallaa et al., 2012; Rice et al., 2015). With an increasing number of the global population residing in urban areas, VE have become the major target for source control in many parts of the world. But 35 uncertainty over the relative importance of diesel and gasoline vehicles to PM$_{vehicle}$ often poses a challenge in effective policymaking (Gertler, 2005).

The Chemical Mass Balance (CMB) Model and Positive Matrix Factorization (PMF) Model are two prevalent receptor models for quantifying contributing sources to $PM_{2.5}$ including VE. However, their capability to resolve separate contributions for diesel and gasoline vehicles is often severely constrained when relying on chemical constituents residing in the PM (particulate matter) fraction alone. In CMB, EC and certain organic compounds (e.g., hopanes, benzo[*ghi*]perylene and coronene) have been specifically used as tracers for deriving diesel and gasoline $PM_{vehicle}$ contributions, respectively (Schauer et al., 1996; Subramanian et al., 2006; Chow et al., 2007). However, the contributions of the two are often subjected to large uncertainty due to the substantial variability in source profiles and oxidation degradation of organic tracers (Subramanian et al., 2006; Weitkamp et al., 2008). On the other hand, PMF analysis of PM compositions often yields one overall VE factor due to the lack of tracers specific to individual VE types and the difficulty is compounded by the often similar temporal pattern of traffic activity among different vehicle types (Dallmann et al., 2014; Lee et al., 2015; Wang et al., 2017). Some other chemical characteristics such as sub-fractions of organic carbon (OC) and EC obtained from thermal analysis and metals (e.g., Mn and Fe) have been used in PMF to differentiate diesel and gasoline contributions (Kelly et al., 2013). These characteristics, however, are relatively less specific and thus they are often not applicable to typical urban areas where a complex mix of contributing sources exists.

In Hong Kong (HK), $PM_{vehicle}$ contribution from diesel vehicles has reduced significantly over the last two decades, resulting from a series of ambitious control efforts. The success has been verified by results from an ad hoc roadside study and a study comparing the emissions in a local tunnel between 2003 and 2015 (Lee et al., 2017; Wang et al., 2018). Most HK studies in the past only reported overall $PM_{vehicle}$ contribution due to the lack of separate local source profiles for diesel and gasoline vehicles and constraints in PMF model posed by a lack of vehicle type specific tracers (Li et al., 2012; Huang et al., 2014; Cheng et al., 2015; Sun et al., 2016). Some studies achieved the separation either by using non-local source profiles in CMB, or by coupling vehicle type specific traffic data collected in a short period with aerosol mass spectrometry-based PMF (Zheng et al., 2006; Lee et al., 2017). The lack of a robust means to differentiate diesel and gasoline contributions to $PM_{vehicle}$ calls for a need to develop a more effective source apportionment strategy, especially considering the long-term need in monitoring their impact on air quality.

In VOC source apportionment studies, certain VOC species have been used to track the contributions of specific vehicle types, such as propane and butanes for vehicles fueled by liquefied petroleum gas (LPG) and pentanes and toluene for gasoline vehicles (Lyu et al., 2016; Yao et al., 2019). These gaseous species, however, are rarely considered for deriving vehicle type specific $PM_{vehicle}$. Lambe et al. (2009) added a few VOC species into their 2-h-resolution organic tracer-based PMF to explicitly apportion black carbon in Pittsburgh to diesel and gasoline vehicles. Thornhill et al. (2010) also used PMF with real-time gaseous species (including VOCs) and $PM_{2.5}$ concentration data captured by a suite of mobile equipment to quantify $PM_{2.5}$ contributions from diesel and gasoline vehicles in Mexico City. Our group reported limited exploration of the combined use of hourly VOCs and $PM_{2.5}$ OC and EC data in PMF to estimate the total $PM_{vehicle}$ at a roadside environment in HK (Huang et al., 2014). Here, we present a detailed investigation on the feasibility of such an approach for separating $PM_{vehicle}$ contributions by diesel and gasoline vehicles using a comprehensive dataset covering six years between 2011 and 2017.

The objective of this work is to establish an approach to obtain vehicle type specific $PM_{vehicle}$, through integrating online routine monitoring data, such as $NO_x$, hourly VOCs and OC-EC measurement data, into PMF analysis. The study features a six-year-long monitoring (2011–2017) in a roadside environment in HK. We evaluate this new method through comparing the total $PM_{vehicle}$ derived from an independent EC-tracer method developed previously, and for the first time report the long-term trends in $PM_{vehicle}$ for diesel and gasoline vehicles in HK (Huang et al., 2014; Wong et al., 2019). Several policies targeting at diesel vehicles fell within the timeline of the study period, providing a valuable opportunity to examine their

effectiveness. The methodology presented in this study for instrument deployment, data collection and analysis could help air quality management authorities to obtain measurement-based evidence from the routine monitoring dataset for evaluating the effectiveness of control policies targeting VE.

## 2 Methods

### 2.1 Roadside Measurements

The study time window spans over a six-year period from May 2011 to August 2017. Sampling was conducted at Mong Kok Air Quality Monitoring Station (MK AQMS), which is a roadside site in the Air Quality Monitoring Network operated by the HK Environmental Protection Department (HKEPD). The station is located at the junction of two trunk roads, with an annual average daily traffic count of ~ 45,000 (Transport Department, 2018). Previous vehicle counting exercise showed that private cars fueled by gasoline, goods vehicles and buses fueled by diesel and taxis running on LPG made up 32 %, 22 %, 16 % and 29 %, respectively, of the vehicle fleet in the sampling area (Lee et al., 2017). In addition to the busy traffic, there are small shops, restaurants, and tall residential and commercial buildings in the immediate vicinity of the station. A previous study has shown that on-road vehicles are the most important source of submicron carbonaceous aerosols at the site, followed by cooking activities (Lee et al., 2015).

A semi-continuous OC-EC field analyzer system (RT-4, Sunset Laboratory, OR, USA) was operated to obtain hourly OC-EC concentration data in $PM_{2.5}$. Details for the on-site operation, instrument conditions and quality control work during the entire sampling period are provided in the supplementary material. Hourly concentration data for $PM_{2.5}$, $C_2$–$C_9$ VOCs, and trace gases including $NO_x$ and CO measured at the MK AQMS were provided by HKEPD. Details of the monitoring equipment can be found in HKEPD's annual air quality report (Environmental Protection Department, 2018). In particular, the VOCs were measured with a GC955 series 611/811 VOC analyzer (Syntech Spectras, The Netherlands) which quantifies 30 species, including 11 $C_2$–$C_8$ alkanes, 9 $C_2$–$C_5$ alkenes, 1 $C_2$ alkyne and 9 $C_6$–$C_9$ single-ring aromatics.

### 2.2 Data Treatment

To avoid biased interpretation of the relationships between the measured species, we excluded hourly samples with one or more missing species from the subsequent data analyses. Sampling days with data cover rate (i.e., number of valid data/total number of hours during the study period) < 75 % were also excluded to maximize the representativeness of the concentration data of a sampling day. The trends presented throughout this study are constructed from the monthly averages. Only months with data cover rate > 33 % are considered. The monthly data cover rates are summarized in Fig. S1.

### 2.3 Estimation of Vehicular $PM_{2.5}$ by Positive Matrix Factorization

In this work, vehicular contributions to $PM_{2.5}$ are quantified by PMF analysis using the EPA PMF 5.0 software (Norris et al., 2014). PMF is a receptor model that solves the chemical mass balance of a speciated sample data matrix by decomposing it into factor profiles and factor contributions with non-negative constraints, with the objective to minimize the objective function Q (Paatero and Tapper, 1994; Paatero, 1997). The Q-value represents the uncertainty weighted deviation between observed and modeled species concentrations.

Hourly concentrations of OC, EC, $NO_x$, CO and 12 selected VOC species from the entire monitoring period are considered in the PMF model for a single analysis. The VOCs, which were consistently detected above detection limit (> 80 % in each calendar year), include ethene, ethane, propane, propene, *i*-butane, *n*-butane, *i*-pentane, *n*-pentane, benzene, toluene,

ethylbenzene and *m-&p*-xylene. Some examples of the excluded VOCs are butadiene, *n*-hexane and *n*-heptane. Preliminary

PMF analysis showed that including these species had no advantage in identifying more sources. Meanwhile, the considerable fraction of below detection limit data for these species would affect the quality of the PMF solutions. Details regarding other modeling inputs are provided in the supplementary material.

Vehicle-related VOCs in the roadside environment are freshly emitted and thus should be mostly conserved, rendering them suitable for receptor modelling. However, for the non-roadside environment, the effect of photochemical reactions should

first be examined, and correction of VOCs input data should be made when needed to avoid bias in source apportioning (He et al., 2019).

$PM_{vehicle}$ for individual vehicle types are calculated as the sum of OM and EC in the corresponding factor profiles, considering it is primarily composed of OM and EC. OM is estimated to be OC multiplying by a factor of 1.2 based on organic aerosol mass spectra measured for both diesel and gasoline VE (Dallmann et al., 2014; Lee et al., 2015).

We note that cooking emissions, a known OC source in MK, is not considered in the current PMF analysis, as measurement of relevant tracer compounds was not supported with the instrumentation deployed in this study. We therefore relax the modeling of OC (i.e., allow the modeled OC to have a relatively larger deviation from measurement) by tripling its uncertainty. This approach allows the apportioning of OC a larger degree of freedom, which in effect allows the model to only capture the OC that is associated with the fitting species, whilst leaving the unrelated fraction as unapportioned OC. A

sensitivity test showed that further doubling the OC uncertainty would not cause a discernible impact on the PMF solution.

The robustness of the PMF solutions is examined through executing the displacement (DISP) and Fpeak (strength values of – 5 and 5) functions. In bootstrap (BS) analysis, the input dataset was split into three groups of equal sample size for execution because of the limited computing capability of the software (total sample size = 24,586). Nevertheless, this practice allows us to assess the model uncertainty associated with using different subsets of samples, which will be discussed in the result

section.

## 2.4 Estimation of Vehicular PM$_{2.5}$ by EC-tracer Method

To evaluate the PMF estimation, an EC-tracer method specifically designed for estimating $PM_{vehicle}$ contribution in roadside environments is applied. Details regarding the principle and application of this method are documented in our previous work and a brief account is given here (Huang et al., 2014; Wong et al., 2019). In this method, VE is assumed to have a

characteristic OC-to-EC ratio ($[OC/EC]_{vehicle}$) and be responsible for all ambient EC. The latter represents a reasonable approximation given that the EC at MK AQMS was dominated by traffic exhaust over the entire study period, which will be discussed further in Sect. 3.1.2. With these assumptions, $OC_{vehicle}$ can be estimated as the product of the ambient EC concentration and the $[OC/EC]_{vehicle}$, while $PM_{vehicle}$ can be estimated using the approach similar to that introduced in Sect. 2.3 (i.e., $PM_{vehicle} = EC_{ambient} + OC_{vehicle} \times 1.2$).

The $[OC/EC]_{vehicle}$ is determined using the minimum OC/EC ratio approach, in which the ambient OC in a certain lowest percentage range by OC/EC ratio are regressed on ambient EC, and the slope obtained represents the target ratio (Lim and Turpin, 2002). This minimum ambient OC/EC ratio is perceived to be of minimal contributions from secondary formation and nonvehicular primary sources, which typically have higher OC/EC ratio than VE (e.g., cooking emissions and biomass burning). In this study, the optimal Deming regression evaluated previously is applied on the lowest 5 % data by OC/EC

ratio on a monthly basis (Huang et al., 2014). The analysis is performed using the Igor (WaveMetrics, Inc. Lake Oswego, OR, USA) based computer program developed by Wu and Yu (2018).

# 3 Results and Discussion

## 3.1 Ambient Trends

### 3.1.1 OC Trends

The monthly OC concentration showed a decreasing trend over the six-year period from May 2011 to August 2017 as shown in Fig. 1a. A consistent seasonal cycle with fall–winter (mid-September to mid-March of next year) high, summer (mid-May to mid-September) low and spring (mid-March to mid-May) in between is observed over the years. The study-wide OC in the three seasonal periods were 6.9±3.4, 3.9±2.6 and 5.9±2.8 µg C m$^{-3}$, respectively (Fig. S2). The main cause for the seasonal variations of OC is related to the geographical location of HK, which is in the coastal area facing the South China

Sea to the south and mainland China to the north. During the fall/winter monsoon season, the prevailing northeasterly wind transports pollutants from the continental area to HK, while in summer the prevailing southerly wind carries clean air masses from the sea (Louie et al., 2005; Hagler et al., 2006; Huang et al., 2014). Another plausible reason for the elevated OC observed in wintertime is the enhanced partitioning of semivolatile organic compounds (SVOCs) into the particle phase due to lower temperature and higher organic aerosol loading. Previous study at the same monitoring site shows that VE-related

organic aerosol (derived from PMF analysis of organic aerosol mass spectra) decreases by 40 % in summer relative to spring despite consistency in traffic flow volume, pointing to a sizable influence of gas-particle partitioning of SVOCs (Lee et al., 2017).

It is noted that the winter OC had larger improvement than summer OC over the monitoring period, as shown in the season-specific trend plot in Fig. S3. The average OC concentration in winter dropped by 6.4 µg C m$^{-3}$ (from 10.7 µg C m$^{-3}$ in 2011

to 4.3 µg C m$^{-3}$ in 2017), while the decrease in summer was 2.3 µg C m$^{-3}$ (from 5.1 to 2.8 µg C m$^{-3}$) during the same period. Such difference demonstrates the benefit on local air quality through collaborative effort in reducing regional air pollution over the years.

### 3.1.2 EC Trends

The six-year trend of EC concentration is plotted in Fig. 1b, which shows a different temporal characteristic compared to

OC. A main feature of the EC trend is the lack of seasonality throughout the years. The study-wide seasonal concentrations remained at ~ 5 µg C m$^{-3}$ in all seasons (Fig. S2). The absence of seasonal variation indicates local emissions dominated EC at this roadside site, and the impact of regional sources on EC, as opposed to OC, was limited. We previously demonstrated that EC at MK AQMS was mainly influenced by vehicular traffic by showing similarities in their diurnal and weekday–holiday variation patterns (Huang et al., 2014). Such correlations persisted over the years, as shown in Fig. S4. Specifically,

during workdays, EC concentration increased 4-fold from its minimum during small hours to ~ 4–8 µg C m$^{-3}$ during daytime. The corresponding increase was 2-fold for the holiday period, consistent with the reduced traffic flow volume. These multiple lines of evidence indicate that EC at the site was mainly affected by local VE sources and less impacted by regional sources.

In the first three years, the monthly EC concentrations fluctuated near the 5–6 µg C m$^{-3}$ range. Starting from mid-2014, it

declined significantly to the level of ~ 3 µg C m$^{-3}$ toward the end of the measurement period. A similar variation trend was also observed for NO$_x$ (Fig. 1c), which is mainly generated by on-road diesel vehicles in the roadside environment. Notably, these decreasing trends coincided with the launch of a Phasing Out Pre-Euro IV Diesel Commercial Vehicles Program in March 2014 in HK. The results here imply diesel vehicles were the major EC contributor at the sampling site.

### 3.1.3 Carbonaceous Aerosols and PM$_{2.5}$ Trends

The relative contributions of carbonaceous aerosols to PM$_{2.5}$ at MK AQMS over the study time window are shown in Fig. 1d. OM was approximated as OC × 1.4 for typical urban aerosols with primary and secondary origins. The PM$_{2.5}$ concentration is overlaid on the same plot. As shown in the figure, PM$_{2.5}$ concentration displayed a seasonal variation (winter high and summer low) similar to that of OC over the years, which was a result of combined effect of regional air pollutant transport and meteorological conditions, as discussed in Sect. 3.1.1. In the middle of the year with warmer weather and lower

PM$_{2.5}$ mass (~ 20 μg m$^{-3}$), EC showed an elevated relative contribution to ~ 30 %, and EC and OM had comparable contributions to PM$_{2.5}$. The opposite was observed in the colder season with higher PM$_{2.5}$ mass (30–60 μg m$^{-3}$). EC only made up ~ 15 % of the aerosol mass while OM accounted for ~ 30 %. About two-thirds of the PM$_{2.5}$ in these periods was composed of non-carbonaceous materials. Based on HKEPD's chemical speciation results for 24-h filter samples, these materials mainly consist of secondary inorganics (sulfate, nitrate and ammonium), followed by crustal material, and trace

elements (Yu and Zhang, 2018). The secondary inorganic components have long been attributed to regional air pollution. The persistently large contributions from these components across the years indicate that controlling PM$_{2.5}$ including its gaseous precursors (SO$_2$, NO$_x$ and NH$_3$) on a regional scale is still important to reduce the overall PM$_{2.5}$ at this roadside location.

### 3.1.4 *n*-Butane and *i*-Pentane Trends

Figure 1 also shows multi-year trends in VOCs that are associated with specific vehicle types (LPG and gasoline). *n*-Butane has been used to track LPG-fueled vehicles at MK AQMS (Lyu et al., 2016; Yao et al., 2019). As shown in Fig. 1e, the *n*-butane level did not show obvious monthly variation over the years, supporting this species was predominantly emitted by local LPG vehicles (box-plot statistics of the monthly concentration are shown in Fig. S5). It remained at ~ 10–15 ppbv level in mid-2011–mid-2013, dropped precipitously to ~ 7 ppbv in the second half of 2013, followed by a steadily declining trend

until the end of the study period. Yao et al. (2019) reported similar trend characteristics for the same site with a more continuous dataset (September 2012–April 2017). The drop in the second half of 2013 was a response to a catalytic converter replacement scheme for LPG-fueled vehicles implemented by the Government (Lyu et al., 2016).

Similar to *n*-butane, *i*-pentane was also dominated by a local source as reflected by the absence of seasonality over the years (Fig. 1f). This gasoline exhaust/evaporation tracer remained fairly stable at ~ 1 ppbv level over the entire study period. Such

an invariability is in line with a previous study showing the VOCs contributions from gasoline-powered vehicles in the same study area were relatively stable over the similar period (Yao et al., 2019).

### 3.2 [OC/EC]$_{vehicle}$ for EC-tracer Method

[OC/EC]$_{vehicle}$ determined using summer months data (June, July and August) are very similar over the years and do not exhibit an obvious trend over the years, as shown in Fig. 2a. The ratios range from 0.30 to 0.47, with R$^2$ between 0.56–0.96

(sample size *n* = 18–33). Figure 2b plots the frequency of occurrence of the lowest 5 % OC/EC ratios in 24 hours of a day for the summer months. It shows that the lowest ratios occurred most frequently near the morning rush hour (7:00–10:00 a.m.) with minimal contribution from other primary sources (e.g., cooking emissions), supporting that these ratios were dominated by VE. The ratios were a factor of 2–3 higher in winter (Fig. S6), likely as a result of enhanced OC contributions from aged air masses and biomass burning from regional transport as discussed in Sect. 3.1.1. Thus, these values were likely biased

high. Another complicating factor is that the reduction in ambient temperature and elevation in organic aerosol concentration in the colder season would favor the partitioning of SVOCs into the particle phase, thereby inflating the [OC/EC]$_{vehicle}$

(Robinson et al., 2007). We did not account for this effect in this study. Instead, the summer values were adopted for deriving the $[OC/EC]_{vehicle}$ and this ratio is considered a lower-estimate for colder season samples. Given that similar $[OC/EC]_{vehicle}$ values were obtained in the summer months across the years, the mean value of 0.35 (standard deviation = 0.05) is considered to be the best estimate of $[OC/EC]_{vehicle}$ for subsequent analysis.

## 3.3 Vehicular Contributions by PMF Analysis

### 3.3.1 Source Identification

Among various PMF solutions, the five-factor solution is the most interpretable for source identification and quantification. The drop in $Q_{true}/Q_{expected}$ value, which reflects the improvement in modeled species concentrations against measurements, is more significant when the factor number is increased by one from three to five compared to from five to eight (Fig. S7). This implies five factors are sufficient and suitable to explain the variations of input species data. The contributing sources are identified through studying the presence of marker species and temporal variations in normalized contributions as shown in Fig. 3.

Three vehicular factors are identified, corresponding to the vehicle types observed near the sampling site. The first factor represents diesel exhaust as it contains the majority of $NO_x$ and EC, which are largely attributed to diesel vehicles. The low OC/EC ratio of 0.5 and diurnal profile, in line with traffic flow of diesel vehicles next to the site, further confirm its source identity.

Factor 2 is associated with gasoline vehicles due to the dominant presence of *i*-/*n*-pentane. This factor also has a diurnal profile consistent with gasoline vehicle flow near the site, with evening peak occurring two hours later than that of diesel vehicles. The *i*-/*n*-pentane could be emitted through fuel evaporation and as unburnt gasoline in tailpipe exhaust, whereas the carbonaceous particulates with OC/EC ratio of 2.5 and CO signal tailpipe exhaust.

Contribution from LPG-fueled vehicles is identified in the third factor by propane and *i*-/*n*-butane originating mainly from fuel evaporation. Its diurnal variation pattern is consistent with the activity pattern (busy in small hours) of local taxis running on LPG (Yu et al., 2016). Note this vehicle type has negligible contribution to PM, in agreement with the highly volatile nature of LPG.

The fourth factor contains notable amount of toluene, ethylbenzene and *m*-&*p*-xylene that commonly exist in consumer and industrial products as solvent or in gasoline as additives (Bolden et al., 2015). Previous studies attributed this factor to solvent usage (Lyu et al., 2016; Yao et al., 2019). However, upon closer examination to its diurnal pattern, we found that this factor shows regular peaks around 11:00 and 17:00. Given the MK AQMS is surrounded by sixteen 24/7 gas stations within 1.5 km (Fig. S8), and the peak business hours of some of these stations show a similar diurnal variation pattern (based on popular times information from Google Maps, Fig. S8), we classify this factor as fuel-filling process instead of solvent usage. The VOCs characteristic ratios (e.g., ethylbenzene-to-*m*-&*p*-xylene ratio) of this profile are also in reasonable agreement with local fuel composition. Details of the comparison are given in Table S3.

The last factor has abundant chemically stable ethane and benzene. Particulates in this factor are also enriched substantially in OC (OC/EC ratio ~ 9). In contrast to the three vehicle-related factors, absence of diurnal variation and presence of winter-high and summer-low contribution were noted (Fig. 3). These characteristics collectively indicate this factor is aged air mass.

### 3.3.2 Model Evaluation

Modeling uncertainty estimation from DISP, Fpeak and BS (grouped samples) shows that the solutions are rotationally and statistically robust, with details provided in the supplementary material. In particular, the factor profiles from the three

grouped PMF analyses are very similar, as shown in Fig. S9. The $[OC/EC]_{vehicle}$ ratios for diesel vehicles are 0.5 in all grouped runs, while those for gasoline vehicles are 1.8–2.2, implying the chemical characteristics of $PM_{vehicle}$ of the two remained similar over the study time period. The PMF solution (base run) is also evaluated from the modeling performance of the fitting species, which are summarized in Table S4. Most gaseous species are well reproduced ($R^2 = 0.73–0.95$), except ethene and ethane ($R^2 = 0.32–0.40$) due to their higher measurement uncertainties. Modeled EC is also consistent with the measurement ($R^2 = 0.90$), but OC in comparison shows larger discrepancy ($R^2 = 0.64$).

To unveil the cause of the OC discrepancy, we compare the modeled and measured OC concentrations on a diurnal basis. The result is presented in Fig. 4 using a box diagram. The plot reveals that the discrepancy mainly occurred during 12:00–14:00 and 18:00–21:00. More specifically, the modeled OC were considerably lower than the measurement during these two periods. This feature remained across all seasons as depicted in Fig. S10. Previous studies have attributed the two organic peaks during these two mealtime periods to cooking emissions (Lee et al., 2015; Sun et al., 2016). As mentioned in Sect. 2.3, we were unable to account for this source in PMF due to the lack of suitable tracer species for cooking emissions. Nevertheless, Fig. 4 shows that the modeled and measured OC had a good agreement during the non-mealtime hours (i.e., 0:00–11:00 and 15:00–17:00). This indicates that the aged air mass factor and VE factors resolved by the PMF were able to explain the non-cooking OC, lending support to the PMF-derived $OC_{vehicle}$. It should be noted that although PMF without down-weighting OC could bring the modeled OC into better agreement with measurement, the resolved factor profiles were less consistent among the three grouped PMF analyses, causing discontinuity in factor contributions between different periods. The improved interpretability of the PMF results indicates that down-weighting OC is necessary in our situation.

### 3.3.3 PMF-derived $OC_{vehicle}$ Trends

The separate diesel and gasoline $OC_{vehicle}$ concentrations over the entire study period are shown in Fig. 5a. A decreasing trend in the overall $OC_{vehicle}$ started to emerge in mid-2014, which was driven by the reduction in diesel $OC_{vehicle}$. Between mid-2011 and mid-2014, diesel $OC_{vehicle}$ hovered at ~ 2 µg C m$^{-3}$ level. It then started decreasing until 2017, upon which the concentration had dropped by half. Gasoline contribution, on the other hand, remained at ~ 1 µg C m$^{-3}$ level over the entire study time period. As a result of the different pace of reductions, gasoline $OC_{vehicle}$ has grown in relative importance over the years as shown in Fig. 5d. In the first three years, gasoline vehicles were a smaller contributor compared to diesel vehicles, accounting for ~ 30 % of the $OC_{vehicle}$, but afterwards, their contribution has become comparable to that of diesel vehicles, growing to 40–50 %. Diesel and gasoline $OC_{vehicle}$ derived from the grouped PMF analyses and the associated uncertainties derived from the 5[th] and 95[th] percentile of the BS results are plotted in Fig. S11a, while the results obtained from the base run are also shown as solid lines in the graph. It appears that the division of samples into three time periods does not exert discernible influence on the results, though the gasoline $OC_{vehicle}$ occasionally deviated from the base result noticeably in the first time period. But in general, $OC_{vehicle}$ from the two vehicle categories have converged over the whole study time period.

The overall $OC_{vehicle}$ derived from the EC-tracer method is plotted in Fig. 5a for comparison. It appears that this independent method can only account for the diesel fraction of PMF-derived $OC_{vehicle}$. A plausible reason is that the $[OC/EC]_{vehicle}$ determined through the minimum OC/EC ratio approach (0.35) is biased toward diesel exhaust with low OC/EC ratio (0.47 from PMF). The comparison here suggests the $OC_{vehicle}$ derived from the EC-tracer method reflects the diesel influence better.

The relative contribution of total $OC_{vehicle}$ to ambient OC exhibited a large seasonal dependence (Fig. S12a). In winter months when ambient OC concentration was high, VE made up 30–40 % of OC. The percentage share increased sharply to 70–100 % in summer months when ambient OC was low. However, it should be noted that the PMF-modeled OC occasionally exceeded the measurement, and the exceedance increased with decreasing ambient OC level (Fig. S13). This

could be attributed to the uncertainties arising from PMF modeling and measuring low level of OC, and thus the relative importance in summer months is likely overestimated.

### 3.3.4 PMF-derived EC$_{vehicle}$ Trends

Trends in vehicular EC (EC$_{vehicle}$) were very similar to those for OC$_{vehicle}$, as shown in Fig. 5b. Different from OC$_{vehicle}$, the EC$_{vehicle}$ derived from the PMF and EC-tracer method, which is essentially the ambient EC, agree well with each other. This is because PMF attributed the majority of the ambient EC to VE sources, more than 70 % as shown in Fig. S12b. As a result, the change in EC$_{vehicle}$ was very similar to that of ambient EC and thus is not repeated here. A key finding is that diesel vehicles dominated the EC$_{vehicle}$ over the entire study period, constituting more than 80 % of EC$_{vehicle}$ as shown in Fig. 5e. Such dominance remains valid after consideration of PMF modeling uncertainties (Fig. S11b). The above findings emphasize the reduction in EC over the years was mostly attributed to the control of diesel vehicles, and this vehicle category should deserve closer attention for further EC abatement.

### 3.3.5 PMF-derived PM$_{vehicle}$ Trends

The positive impact on air quality of vehicle control policies, if any, is more obvious if EC and OM from VE are considered together (i.e., PM$_{vehicle}$). The monthly average PM$_{vehicle}$ by vehicle category are given in Fig. 5c. As of mid-2014, PM$_{vehicle}$ fluctuated slightly around the 8 $\mu$g m$^{-3}$ level, followed by a considerable reduction to ~ 4 $\mu$g m$^{-3}$ level in the beginning of 2017. This amount of reduction represents one-fifth of typical PM$_{2.5}$ concentration at MK AQMS during summer (~ 20 $\mu$g m$^{-3}$). As noted from Fig. 5c, the reduction in PM$_{vehicle}$ was mainly driven by diesel vehicles, which was also the dominant PM$_{vehicle}$ contributor over the whole study period as given in Fig. 5f. Another finding from Fig. 5f is that the relative importance of gasoline vehicles has only grown slightly from ~ 20 % before mid-2014 to ~ 30 % afterwards despite the drastic decrease in diesel PM$_{vehicle}$. After considering the PMF modeling uncertainties as depicted in Fig. S11c, it is still clear that diesel vehicles have dominated PM$_{vehicle}$ at MK AQMS over the entire study period, and this vehicle class should remain the focal point for further control of PM$_{vehicle}$.

PM$_{vehicle}$ estimated by the EC-tracer method are also compared with the PMF method in Fig. 5c. As shown, the disparity between the two approaches has narrowed considerably compared to OC$_{vehicle}$. The improvement mainly involves the consideration of EC, which is similarly perceived as tracer for VE in the two estimation methods. The similarity between the PM$_{vehicle}$ obtained from the two methods lends support to our PMF model in producing reasonable estimates for separate diesel and gasoline PM$_{vehicle}$ contributions. Furthermore, the consistent results obtained from both methods highlight that PM$_{vehicle}$ was an important contributor to the PM$_{2.5}$ at MK AQMS, with noticeable seasonal variation similar to that of OC$_{vehicle}$, as given in Fig. S12c. Based on the PMF results, during summer and under the dominance of local sources, VE was responsible for ~ 30–60 % of ambient PM$_{2.5}$ during the study period. When PM$_{2.5}$ concentration increased in winter due to regional influence, VE contribution dropped to roughly 10–20 %.

### 3.4 Policy Evaluation

Tackling tailpipe emissions from diesel commercial vehicles (DCVs) and franchised buses is a long-term need in HK (Environment Bureau, 2013). A three-day detailed traffic counting exercise was conducted by the Government at the MK AQMS in May 2013. The details were reported in Lee et al. (2017). From that we are able to identify a truck-dominated period around midday (11:00–13:00) during which the number of trucks (i.e., DCVs) is a factor of 2–3 higher than that of buses (~ 100 vs. ~ 40 vehicles h$^{-1}$), as well as a mid-night period (22:00–0:00) with buses count being 3 times higher than

the count of trucks ($\sim 60$ vs. $\sim 20$ vehicles h$^{-1}$). Hence, the annual trends in diesel PM$_{vehicle}$ extracted from these two periods could provide indication on how diesel PM$_{vehicle}$ was impacted by different policies. The results are presented in Fig. 6. Also shown in the figure is a background condition represented by the small hours (2:00–4:00) when diesel vehicle number reaches the minimum ($\sim 10$ vehicles h$^{-1}$). A clear declining trend in diesel PM$_{vehicle}$ is noted for both vehicle-dominated periods, with both trends approaching the levels in the background period.

For the truck-dominated trend, the PM$_{vehicle}$ levels started to drop after March 2014, which marks the commencement of the Phasing Out Pre-Euro IV DCVs Program implemented by the Government. This program aims at progressively replacing all Pre-Euro IV DCVs ($\sim 82,000$) in HK by the end of 2019 (Environment Bureau, 2013). The diesel PM$_{vehicle}$ concentrations in the pre-DCV Program period and the start of 2017 were respectively $\sim 9$ and $\sim 5$ μg m$^{-3}$, representing almost a 50 % reduction. The reduction also appeared to respond reasonably with the progress of the program shown in Fig. 6. Prior to this DCV program, another scheme that replaces $\sim 7,400$ Euro II DCVs was launched during July 2010–June 2013. That program, however, did not produce an obvious impact on diesel PM$_{vehicle}$, possibly because of the smaller scale of the implementation compared to the more recent DCV program.

The declining trend in diesel PM$_{vehicle}$ during the bus-dominated period could be attributed to a host of control measures for franchised buses. In the intervening years, local franchised bus companies have been continuously scrapping and replacing old model buses with newer model buses complying with higher Euro Standards (Environment Bureau, 2013). At the level of transportation management, the Government has been pursuing reduction of bus trips in congested corridors through rationalization of bus routes. Setting up of low emission zones in densely populated spots including the MK area, where only cleaner model franchised buses are allowed, might also contribute to the decrease in bus-related PM$_{vehicle}$ contribution (Environment Bureau, 2013).

## 4 Conclusions

We present a holistic analysis on the long-term monitoring data of hourly PM$_{2.5}$ OC and EC, vehicle-specific VOCs (e.g., *n*-butane and *i*-pentane) and NO$_x$ concentrations in an urban roadside environment in HK. The dataset covers a six-year period from May 2011 to August 2017. Both OC and EC concentrations were observed to decrease notably over the entire study period, plausibly due to the efficient control of pollution sources on both regional and local context. By integrating OC, EC and VOCs (e.g., *n*-butane, *i*-pentane, benzene and xylene) datasets into PMF analysis, we successfully differentiate PM$_{vehicle}$ contributions from diesel and gasoline vehicles, and for the first time report their individual long-term trends. The overall PM$_{vehicle}$ is also estimated by the EC-tracer method, which shows good agreement with that from the PMF analysis, supporting the PM$_{vehicle}$ estimate from the PMF. Our work identifies diesel vehicles as the dominant vehicle type in contributing PM$_{vehicle}$ ($\sim 70$–80 %) over the entire study period. Thus, further VE control effort for mitigating roadside PM$_{vehicle}$ in HK should focus on diesel vehicles. The technique developed in this work could be extrapolated to other roadside environments with mixed vehicular contributions, considering both continuous OC and EC analyzers and online VOC instruments are increasingly incorporated in governments' air quality monitoring programs. We note that the OC$_{vehicle}$ estimated by this approach serves as a lower limit for the vehicle-contributed OC since the fitting species considered here are all tracers for primary emissions. The primary emissions from on-road vehicles also have high potential to form secondary organic aerosols (Gentner et al., 2017), and this secondary PM derived from vehicles is not captured in PM$_{vehicle}$ estimates by our method. Future work should attempt to quantify this missing fraction of vehicular PM$_{2.5}$ for more insightful policy implications.

*Data availability.* The data presented in this article are available from the authors upon request. Please contact Jian Zhen Yu (jian.yu@ust.hk).

*Author Contribution.* JZY, XHHH, and YKW designed the field experiments with assistance from PKKL, ALCY and DHLC. YKW and XHHH carried out the ECOC measurements. YKW performed the data analysis and prepared the manuscript with contributions from all co-authors.

*Competing interests.* The authors declare that they have no conflict of interest.

*Disclaimer.* The content of this paper does not necessarily reflect the views and policies of the HKSAR Government, nor does mention of trade names or commercial products constitute an endorsement or recommendation of their use.

*Acknowledgements.* This work is supported by the Hong Kong Environmental Protection Department (HKEPD) (tender refs. AS10-336, 12-00255, and 14-05593) and Hong Kong Research Grants Council (16305418). We thank HKEPD for provision of the hourly VOC dataset.

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

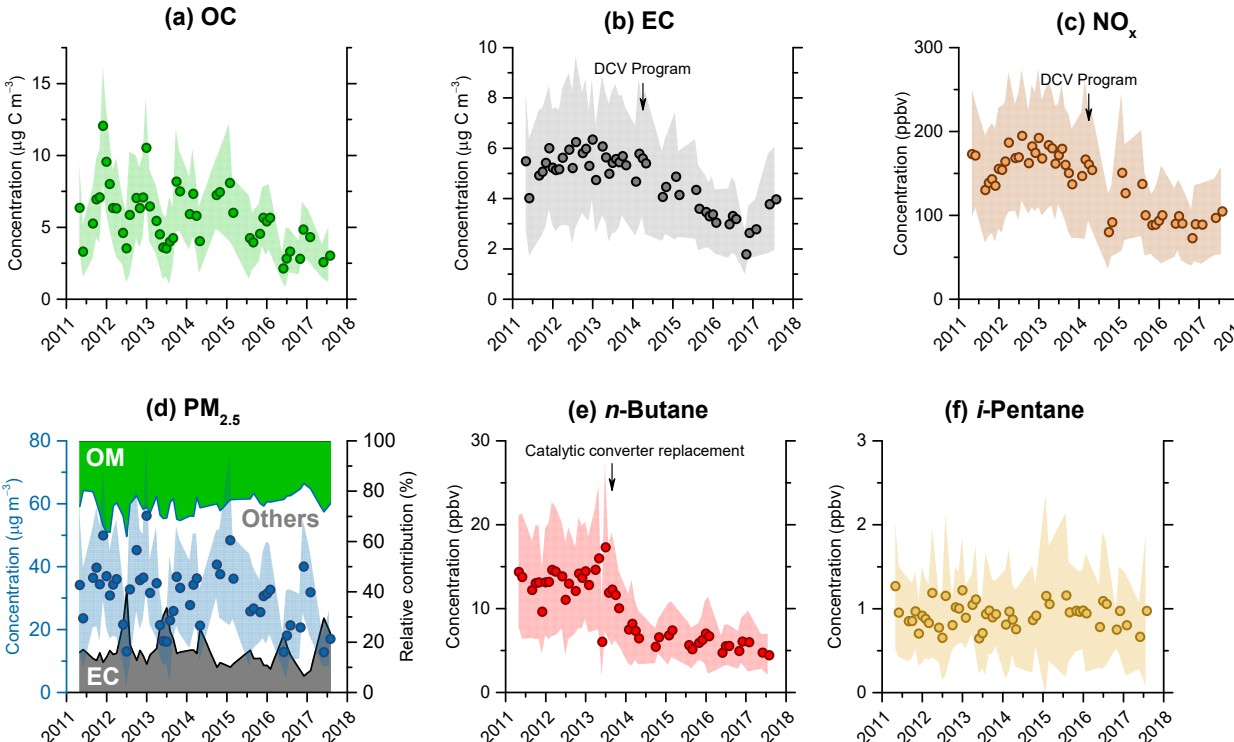

Figure 1. Trends in concentrations of (a) OC, (b) EC, (c) NO$_x$, (d) PM$_{2.5}$, (e) *n*-butane and (f) *i*-pentane at MK AQMS. Each data point represents the monthly average of the hourly concentrations. Shaded areas represent one standard deviation for the hourly 510 data. Commencements of the Phasing Out Pre-Euro IV Diesel Commercial Vehicles (DCV) Program and Catalytic Converters Replacement Program for LPG taxis are marked in figure (b), (c) and (e). In figure (d), relative contributions to PM$_{2.5}$ from EC, OM and other components are shown in secondary axis. OM is approximated as OC × 1.4. Other components are the difference between measured PM$_{2.5}$ and EC + OM.

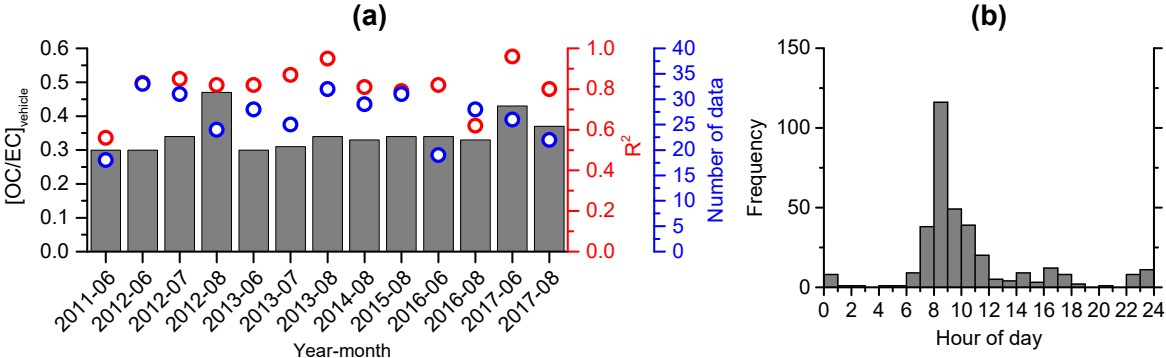

Figure 2. Determination of [OC/EC]$_{vehicle}$ for MK AQMS using optimal Deming regression of the lowest 5 % data by OC/EC ratio from summer months. Figure (a) presents the month-by-month [OC/EC]$_{vehicle}$ (grey columns), $R^2$ of OC and EC (red markers) and number of data point considered (blue markers). Figure (b) shows the frequency of occurrence of the lowest 5 % OC/EC ratios at different hours of the day considering all of the summer data.

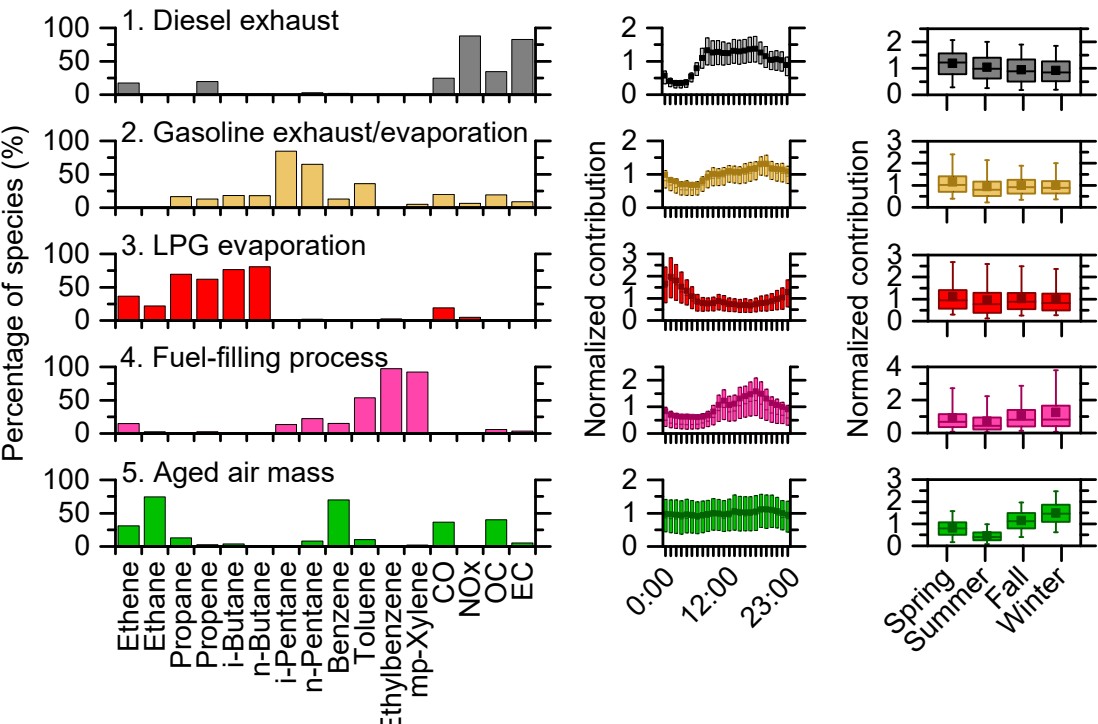

**Figure 3. Factor profiles resolved by PMF analysis of hourly PM$_{2.5}$ OC-EC, C$_2$–C$_8$ VOCs and other trace gases data at MK AQMS. Panels on the left present the species relative contributions to each factor. Middle and right panels respectively show the diurnal and seasonal profiles of normalized contribution for each factor in box-plots. For each box, solid square marker, horizontal line, lower and upper bound are mean, median, 25$^{th}$ and 75$^{th}$ percentile, respectively. Whiskers in the seasonal plots represent 5$^{th}$ and 95$^{th}$ percentile.**

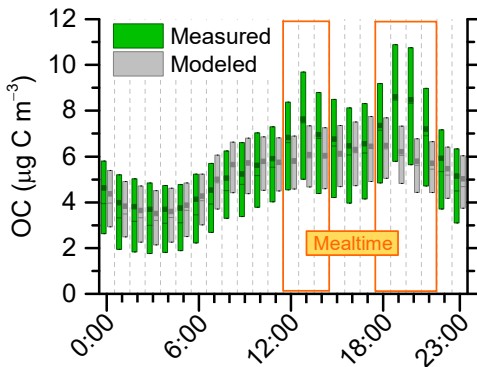

**Figure 4. Diurnal variations of OC concentrations in box-plots derived from ambient measurement (green boxes) and PMF modeling (grey boxes). For each box, solid square marker, horizontal line, lower and upper bound are mean, median, 25$^{th}$ and 75$^{th}$ percentile, respectively. The afternoon and evening mealtime periods are indicated in the orange frames.**

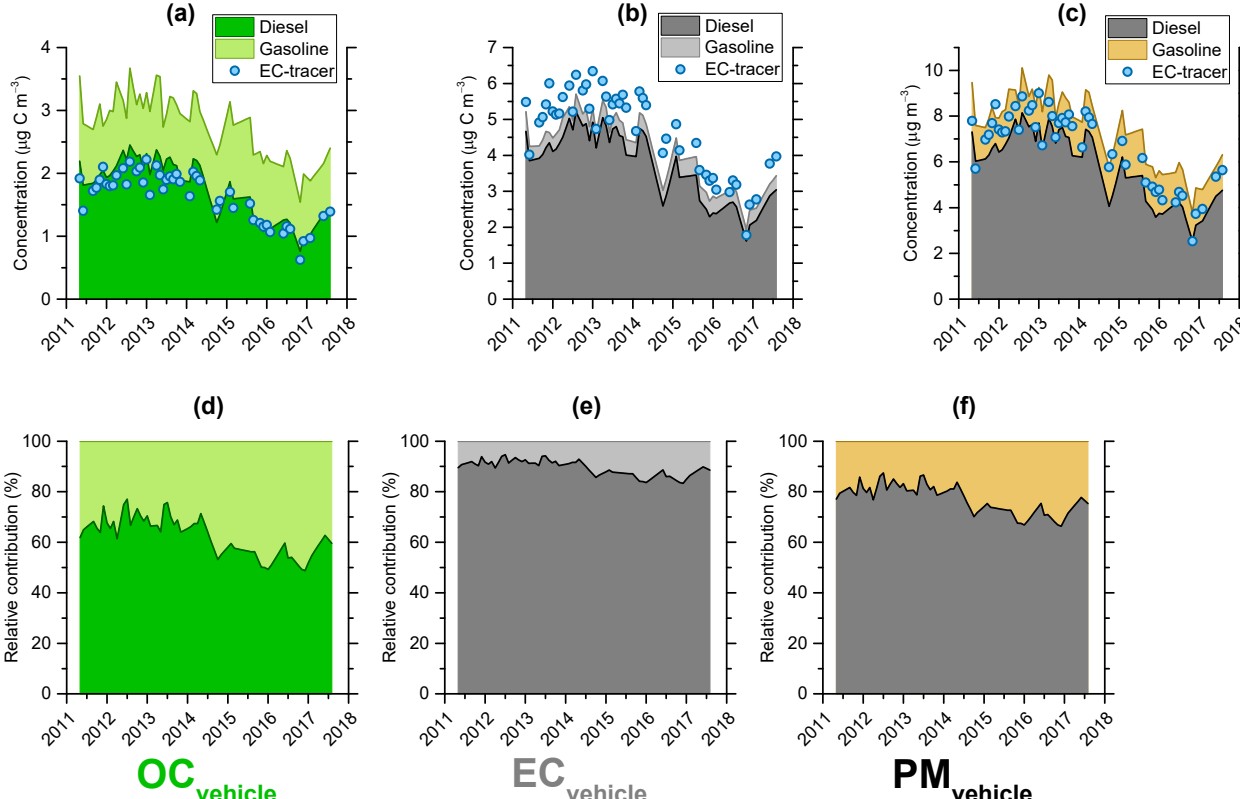

**Figure 5. Trends in contributions of diesel and gasoline vehicles at MK AQMS derived from PMF analysis (stacked areas) in terms of (a) OC_vehicle, (b) EC_vehicle and (c) PM_vehicle concentrations. Relative contributions of the two vehicle types to the corresponding pollutants are shown in figure (d)–(f). The stacked areas are constructed by interpolation of the monthly data points. In all plots, each data point represents the monthly average of the hourly concentrations. Only months with data cover rate > 33 % are considered. Overall vehicular contributions derived from the EC-tracer method are shown in blue markers in figure (a)–(c).**

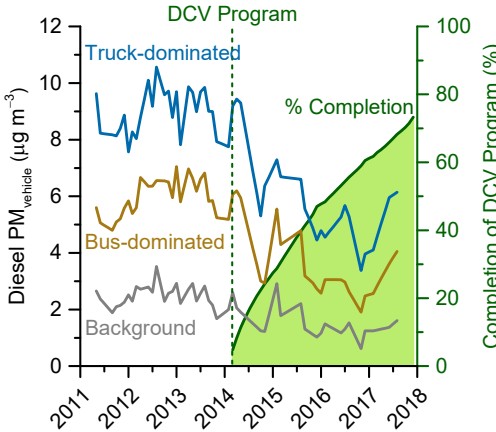

**Figure 6. Trends in PMF-resolved diesel PM_vehicle concentrations (left axis) at the MK AQMS during truck-dominated hours (11:00–13:00, blue line), bus-dominated hours (22:00–0:00, brown line), and background hours (2:00–4:00, grey line). The vertical dashed line represents commencement of the Phasing Out Pre-Euro IV Diesel Commercial Vehicles Program (in March 2014). The green-shaded area represents the percent completion of the program (right axis).**