# Peer review of "Tracking Separate Contributions of Diesel and Gasoline Vehicles to Roadside PM2.5 Through Online Monitoring of Volatile Organic Compounds, PM2.5 Organic and Elemental Carbon: A Six-Year Study in Hong Kong"

_Atmospheric Chemistry and Physics, 2020_

## Referee Comment (RC1) · Anonymous Referee #2 · 3 Apr 2020

General Comments:

In this work, Wong et al. developed and validated a positive matrix factorization (PMF)-based analysis approach to apportion PM2.5 organic carbon (OC) and elemental carbon (EC) measurements into specific vehicle type source contributions. Authors conducted PMF analysis using air quality monitoring data collected over a 6-year measurement period at a near road site in Hong Kong that included semi-continuous OC

and EC, NOx, CO, and speciated volatile organic compounds. Vehicular contributions to OC and EC attributed to diesel and gasoline vehicle exhaust along with three other factors were calculated by PMF analysis. Seasonal and diurnal trends of these source contributions as well as the ambient measurements were discussed in detail. The manuscript was clearly written, and methods were sufficiently explained.

The PMF analysis required only 16 inputs from measurements that are routinely conducted at numerous ambient air monitoring networks, so this apportionment method could potentially be widely used by regulatory agencies. However, one major issue with this manuscript is that authors have not properly explained limitations of this method – can this approach be used in other air monitoring conditions where vehicle emissions are not the major source of PM or other locations outside of Hong Kong? Is the method only robust for data collected from near road sites in Hong Kong? Further guidance on caveats regarding application of this approach to other situations/locations is needed. The other related issue is that the method is heavily weighted towards primary vehicle emissions without accounting for temperature-dependent semivolatile organic partitioning to particle phase or including good tracers for regional air pollution or primary emissions from any other source. Therefore, it's not clear whether trends in OC seasonality are due to SVOC partitioning or regional transport. Further discussion of how to better address these method limitations (e.g. potential underestimation of vehicle related primary/secondary OC or inaccurate contribution of regional air pollution) to better inform regulatory action would be helpful. It is recommended that after addressing these issues, the manuscript may be considered for publication with minor revisions.

Specific comments: Lines 78-79. Clarify what is meant by "highly recommended". Line 156. Is there a prevalent wind direction in other seasons? Line 161. The results presented here do not appear to explain these reductions in OC over time. Are there any suggested explanations? Could it be improvements in regional air pollution? Line 176. If vehicle exhaust is the main source of OC, why not use OC x 1.2 to estimate OM as

was used with the EC Tracer Method? Lines 183-187. Could these measurements be used in this PMF analysis as a tracer(s) for improve regional air pollution factor? Line 197. Do benzene and ethane have seasonality from their contribution in aged air mass? Line 209. Could average ambient temperatures be plotted over this time period in the supplementary material so that partitioning could potentially be estimated? Line 214. How does this value compare to those determined in vehicle emissions studies or in emission inventories for representative vehicles in Hong Kong? Line 236. Are the relative VOC contributions in the fuel filling profile consistent with the local gasoline fuel composition? Line 274. Is there a possible explanation for this deviation? Line 280. If OC from other factors are added, does the seasonality disappear? Line 305. This would be a good point to stress that limitations of this method in accurately estimating PMvehicle (does not take SOA formation, SVOC partitioning, or other emission sources into account) makes it difficult to make these types of policy recommendations. Line 334. The trends for the truck and bus time periods looks strikingly similar. Was this expected? Perhaps traffic patterns have changed since the traffic counting exercise? Figure 1. What is the air quality PM2.5 standard level? Figure 3. It would be helpful to add Fig. S6 to this figure. Figure S10. It would be useful to see other factor contributions. Figure S11. A time series of modelled and measured OC would be helpful to include.

---

## Author Comment (AC1) · 5 May 2020

Point-by-Point Response to Review Comments by Anonymous Referee #2 on "Tracking Separate Contributions of Diesel and Gasoline Vehicles to Roadside PM2.5 Through Online Monitoring of Volatile Organic Compounds, PM2.5 Organic and Elemental Carbon: A Six-Year Study in Hong Kong" by Yee Ka Wong et al.

**Response to General Comments:**

We thank the reviewer for the comments, which have helped us sharpen our understanding of the limitations of current work and the way forward in improving the accuracy of estimating vehicular emission contributions using PMF model. Each of the questions raised by the reviewer has been addressed and detailed in the ensuing point-by-point response. We have addressed all the other comments as well and offered detailed explanation where we disagree with the reviewers. Our response text is marked in blue in this document. The revised text in the main manuscript is also marked in blue. References cited in this response document are placed at the end.

"However, one major issue with this manuscript is that authors have not properly explained limitations of this method – can this approach be used in other air monitoring conditions where vehicle emissions are not the major source of PM or other locations outside of Hong Kong? Is the method only robust for data collected from near road sites in Hong Kong? Further guidance on caveats regarding application of this approach to other situations/locations is needed."

**Response:** In principle this PMF approach is applicable in places where vehicle emissions are not the major PM source. As long as vehicle-related tracer compounds are available (e.g. pentanes and EC), the vehicular contributions could be teased out using PMF model. As to whether the method is constrained to roadside environment (in Hong Kong), we opine that the method gives the most robust results in near roadside site condition where the speciated VOCs are freshly emitted and are less affected by photochemical degradation, as mentioned in line 117 of the original main text. However, the reactivity of VOCs should be taken care of when applying the method in area downwind of traffic emissions source region.

The following statement will be added to the revised main text:

Line 118: "However, for non-roadside environments, the effect of photochemical reactions should first be examined, and correction of VOCs input data should be made when needed to avoid bias in source apportioning (He et al., 2019)."

"The other related issue is that the method is heavily weighted towards primary vehicle emissions without accounting for temperature-dependent semivolatile organic partitioning to particle phase or including good tracers for regional air pollution or primary emissions from any other source. Therefore, it's not clear whether trends in OC seasonality are due to SVOC partitioning or regional transport. Further discussion of how to better address these method limitations (e.g. potential underestimation of vehicle related primary/secondary OC or inaccurate contribution of regional air pollution) to better inform regulatory action would be helpful."

**Response:** We agree that differentiating the effects of SVOC partitioning and regional transport on OC seasonality could be very useful for shaping policy development. Attaining this goal, however, is challenging at this stage due to constraints in available speciation data. Sulfate and water-soluble potassium have been demonstrated to be useful for tracking regional secondary aerosol formation and biomass burning, respectively, in our study region (e.g. Hu et al., 2010; Huang et al., 2014). We believe deployment of

additional monitoring system for online measurement of ionic and elemental compositions in the future would effectively allow us to gain a more comprehensive picture on the sources affecting the study region.

We also agree that SVOC partitioning would affect vehicular OC contributions. A possible means to examine this issue is through conducting the PMF analysis on subset of samples grouped by temperature and organic aerosol concentration. The effectiveness of this undertaking requires further investigation. Indeed, while this response document is being drafted, we are preparing another manuscript which incorporates gas-particle partitioning of organic aerosol and oxidation degradation of hopanes (molecular source tracers for vehicular emissions) into chemical mass balance model to quantify vehicular contributions to PM2.5. We believe the study would provide further insights into the effect of atmospheric processes on PMF quantification of vehicular contributions. We have addressed the study limitations issue in the ensuing specific comments section.

**Response to Specific Comments:**

Lines 78-79. Clarify what is meant by "highly recommended".

**Response:** Revision is made as below for clarification:

Line 78: The methodology presented in this study for instrument deployment, data collection and analysis could help air quality management authorities to obtain measurement-based evidence from the routine monitoring dataset for evaluating effectiveness of control policies targeting VE.

Line 156. Is there a prevalent wind direction in other seasons?

**Response:** Yes, in summer the prevailing wind is from the south. Revision is made as follow:

Line 158: During the fall/winter monsoon season, the prevailing northeasterly wind transports pollutants from the continental area to HK, while in summer the prevailing southerly wind carries clean air mass from the sea (Hagler et al., 2006).

Line 161. The results presented here do not appear to explain these reductions in OC over time. Are there any suggested explanations? Could it be improvements in regional air pollution?

**Response**: Yes, we agree that the improvement especially during winter was largely due to reduced regional air pollution. Revision is made as follow:

Line 162: It is noted that the winter OC had larger improvement than summer OC over the monitoring period. The average OC concentration in winter dropped by 6.4  $\mu$ g C m-3 (from 10.7  $\mu$ g C m-3 in 2011 to 4.3  $\mu$ g C m-3 in 2017), while the decrease in summer was 2.3  $\mu$ g C m-3 (from 5.1 to 2.8  $\mu$ g C m-3) during the same period. Such difference demonstrates the benefit on local air quality through collaborative effort in reducing regional air pollution over the years.

Line 176. If vehicle exhaust is the main source of OC, why not use OC x 1.2 to estimate OM as was used with the EC Tracer Method?

**Response:** Though vehicle exhaust represents an important OC source at the site, the contributions from aged air mass and cooking emissions are non-negligible, which are more oxygenated and thus having higher OM/OC ratio. We therefore use 1.4 to estimate the overall OM.

Lines 183-187. Could these measurements be used in this PMF analysis as a tracer(s) for improve regional air pollution factor?

**Response:** The chemical composition data adopted here is from 24-h filter-based measurement, and thus are not compatible with the hourly data set for PMF analysis. Deployment of MARGA system for hourly monitoring of ionic species (e.g. sulfate, nitrate,  $K^+$ , etc) could expand the source apportionment capability of this site. Revision is made for clarification:

Line 187: Based on HKEPD's chemical speciation results for 24-h filter samples, these materials mainly consist of secondary inorganics (sulfate, nitrate and ammonium), followed by crustal material, and trace elements (Yu and Zhang, 2018).

Line 197. Do benzene and ethane have seasonality from their contribution in aged air mass?

**Response**: Yes, the two species showed seasonality in line with contribution of aged air mass, as shown in Figure R1 below. We attempt to focus on vehicle-related species and thus these VOCs were not discussed in detail in the main text.

**Figure R1.** Trends in concentrations of (a) ethane and (b) benzene at MK AQMS. Each data point represents the monthly average of the hourly concentrations. Shaded areas represent one standard deviation for the hourly data.

Line 209. Could average ambient temperatures be plotted over this time period in the supplementary material so that partitioning could potentially be estimated?

**Response**: The average ambient temperatures are plotted in Figure R2. As shown in the figure, the temperature varied in a fairly constant manner over the years, which suggests the effect of temperature on partitioning should be largely consistent over the study period. This figure will be included in the supplementary material in our next revision.

**Figure R2.** Trend in ambient temperature near MK AQMS. Each data point represents the monthly average of hourly data. Shaded area represents one standard deviation for the hourly data.

Line 214. How does this value compare to those determined in vehicle emissions studies or in emission inventories for representative vehicles in Hong Kong?

**Response:** Wang et al. (2018) reported that the OC/EC ratio measured from a tunnel in Hong Kong in 2015 winter was  $0.7\pm0.2$ , which is higher than the [OC/EC]vehicle ratio determined by the minimum OC/EC ratio method in our study (i.e.,  $0.35\pm0.05$ ). The higher ratio observed in Wang et al.'s work could potentially be attributed to lower temperature (measurement taken in winter vs. summer data adopted in our study) and the higher organic aerosol loading in tunnel (OC concentration was ~17 µg C m-3 in the tunnel vs. < 5 µg C m-3 at MK AQMS during summer), which favor the partitioning of SVOC into particle phase. The difference in prevailing driving conditions between the two sites may also contribute to the OC/EC ratio discrepancy. Vehicle emissions at our roadside site are more influenced by emissions from engine acceleration, due to the close proximity to several pedestrian crossings, which in turn gives higher EC emission compared to tunnel environment where vehicles are mainly traveling at constant speed (Lee et al., 2017).

Line 236. Are the relative VOC contributions in the fuel filling profile consistent with the local gasoline fuel composition?

**Response:** The relative VOC contributions in the fuel filling profile are in reasonable agreement with local gasoline and diesel fuel composition reported by Tsai et al. (2006). We will add Table S3 in the supplementary information for illustration and revise the main text as below.

Line 241: The VOCs characteristic ratios (e.g. ethylbenzene-to-*m*-&*p*-xylene ratio) of this profile are also in reasonable agreement with local fuel composition. Details of the comparison are given in Table S3.

**Table S3.** Comparison of VOCs characteristic ratios among fuel-filling process profile derived in this study and gasoline and diesel fuel profiles reported in Tsai et al. (2006)

|                                       | Fuel-filling process | Gasoline fuel      | Diesel fuel         |
|---------------------------------------|----------------------|--------------------|---------------------|
|                                       | (This study)         | (Tsai et al. 2006) | (Tsai et al., 2006) |
| Toluene/Benzene                       | 12.4                 | 22.1±12.7          | ~10*                |
| Toluene/ m -& p -Xylene | 1.8                  | 6.3                | ~1*                 |
| Ethylbenzene/m-&p-Xylene              | 0.5                  | ~0.5*              | ~0.25*              |

Note: The value is approximated from graphic information as no numerical values were provided in the publication.

Line 274. Is there a possible explanation for this deviation?

**Response:** We double check the result and confirm there is no erroneous data presentation. We compared the modeling performance of *i*-pentane for the two sets of PMF analyses and did not found any significant discrepancy, as shown in Figure R3, indicating the discrepancy was caused by other unknown reasons. Indeed, the apportioning of OC by the PMF method in this study is affected by a complex web of factors, including PMF modeling uncertainties, partitioning of SVOCs, and omission of source tracers of certain known sources (e.g. cooking emissions). Further work is needed to better constrain the vehicular OC contributions (both primary and secondary OC). Nevertheless, this work highlights that when both OC and EC are considered, diesel vehicles are the more important source of primary vehicular PM2.5 than gasoline vehicles in the study area.

**Figure R3.** Scatter plot comparison of modeled versus measured *i*-pentane derived from (a) base PMF analysis and (b) PMF analysis with grouping.

Line 280. If OC from other factors are added, does the seasonality disappear?

**Response:** Figure S10 is re-plotted as Figure R4 in this response document by adding OC contributions from fuel-filling process and aged air mass factors (contribution from LPG vehicles are zero). As shown in the figure, the seasonality becomes much less significant after OC from other factors are added.

---

## Referee Comment (RC2) · Anonymous Referee #1 · 24 May 2020

General comments:

This manuscript presented source apportionment of carbonaceous aerosols based on six-year in situ measurements of different types of gas- and particle-phase chemical species at a near road-side site in Hongkong. Specially, the data sets of organic and elemental carbon coupled with VOCs were applied in PMF analysis to apportion the source contributions of gasoline and diesel combustion emissions. Meanwhile, an EC-

tracer was also applied here for the comparison with the PMF results. Both methods have confirmed a significant contribution of traffic emissions to ambient aerosols in such an urban environment. A decreasing trend in diesel-related emissions was observed, which was attributed to local emission control policies. Overall, this paper is well written and easy to follow, as well as it fits well this journal scope. I believe that this work could make significant implication for understanding the relationship between emission control and air quality improvement, as a good example in China. I would suggest this paper could be accepted for publication before addressing the comments below.

Comments in detail:

Line 128: More description/discussion on PMF analysis should be given. For example, did the authors performed seasonal PMF runs, or only yearly PMF runs, or only a single PMF run (six-year data all together)? Did the authors test more factors? Why not selecting more factors? How were those Q value variations?

Line 160, section 2.4: the author defined that the vehicle PM is the sum of total ambient EC and vehicle OC. Was there contribution of any solid-fuel burning (and cooking to OC) to EC and OC? If yes, how did the authors isolate these fractions using this vehicle OC/EC method, and assuming that total ambient EC was only from vehicle emissions? In addition, the authors simply considered the minimum OC/EC ratio as the vehicle OC/EC method. This part should be more clearly described, for example what this ratio is, how many selected samples associated with such minimum ratio, did the authors filter the potential influence of biomass burning and/or cooking (if applicable), etc.? These points should be discussed here.

Line 180, section 3.1: For trend analysis/discussion in the manuscript, they could be further performed with statistical approach, for instance, the Mann-Kendall trend test. By using that, the magnitude of change rate for the trend can be quantified, along with the significance levels. Overall, it seems there were increasing trends during the beginning years, while the decreasing trends have been observed since around 2013? In addition, it seems that there was a rapid decrease in those trends during around 2013-2015 (?), while a slowdown decrease in those trends were observed during recent years. Therefore, those points might be further discussed and explain possible reasons.

Lines 186 – 190: I think air mass back-trajectory analysis could be applied here to support those statements. By this method, the authors will be able to investigate the different concentration levels and/or sources of OC and EC associated with different air mass origins/clusters.

Lines 190 – 192: Did the authors have any evidence to prove these reasons?

Lines 192 – 193: It could help to further support the OC trend driven by wintertime OC when you separately show the six-year trends of monthly data for winter (DJF), spring (MAM), summer (JJA), and fall (SON). Did the authors find seasonal characterization for those OC and EC trends over the six-year period? Those new plots can be presented in supplement.

Lines 202-203: Based on these discussions above, it seems not fully convinced to conclude the less regional source influence on EC loadings rather than local traffic emissions. I guess, the similar EC diurnal cycles between work days and holidays/Sundays might reflect similar rush hours between the two types of days during a week. This could not sufficiently prove that the EC was more coming from local emissions. The different concentration levels of EC between the two types of days were also observed, however they weren't discussed. These similar diurnal patterns, along with different concentration levels, would be due to reduction in the total amount of traffic emissions over local and/or small-regional scales, however rush hours were overall not changed. In addition, as the NOx data was available in this work, it would be interesting to show correlations of EC versus NOx during work days and holidays, respectively. As commented above, air mass back trajectory analysis could also help to understand if there

would have significant influence of regional sources on EC observed at the receptor site. Therefore, further discussions to support your statements should be extended.

Line 241: It's not easy to justify the seasonality only based on time series of monthly data. It would be better if the authors could show monthly cycles and/or perform a seasonality significance test.

Lines 257-259: It would be also good to show diurnal variations of the OC/EC ratio to support the lowest ratios associated with the rush hours. As shown in Fig. S3, EC presents high concentration starting from around 7 AM – 6 PM. Could this suggest rush hours for EC spanning this time period? It might be also useful to check and discuss diurnal variations of OC concentrations, NOx and OC/EC ratios.

Line 278: As commented above, did the authors perform only a single PMF run? Did you have any other PMF run tests, e.g., using seasonal runs? Based on these runs, did the authors have the same solution? And were the results from seasonal PMF runs consistent with the present results? Did the authors try to increase the number of PMF factors? How were those PMF solutions based these tests?

---

## Author Comment (AC2) · 19 Jun 2020

***Point-by-Point Response to Review Comments by Anonymous Referee #1 on "Tracking Separate Contributions of Diesel and Gasoline Vehicles to Roadside PM$_{2.5}$ Through Online Monitoring of Volatile Organic Compounds, PM$_{2.5}$ Organic and Elemental Carbon: A Six-Year Study in Hong Kong" by Yee Ka Wong et al.***

**Response to General Comments:**

We thank the reviewer for the comments and suggestions, which have helped us to supplement and consolidate our research findings. Each of the questions raised by the reviewer has been addressed and detailed in the ensuing point-by-point response. We have addressed all the other comments as well and offered detailed explanation where we disagree with the reviewers. Our response text is marked in blue in this document. The revised text in the main manuscript is also marked in blue. References cited in this response document are placed at the end. For the ease of referencing, the line numbers mentioned in our response refer to those in the revised main text/SI documents.

**Response to Specific Comments:**

Line 128: More description/discussion on PMF analysis should be given. For example, did the authors performed seasonal PMF runs, or only yearly PMF runs, or only a single PMF run (six-year data all together)? Did the authors test more factors? Why not selecting more factors? How were those Q value variations?

**Response:** We thank the reviewer for reminding us to provide additional necessary details on the PMF analysis. The PMF-derived source contributions reported in this work is based on a single PMF run considering six-year data all together. For the sake of bootstrap uncertainty estimation, we also divided the whole dataset into three time-segments, each segment consists of one-third of the samples, and performed PMF for each time segment. This exercise and the relevant outcome have been discussed in line 132–135, and line 262–266, respectively.

As to the number of factors determined in the final solution, after considering the reviewer's comment, we added the Q-value metric (i.e., $Q_{true}/Q_{expected}$), which reflects the discrepancy between modeled and observed species concentrations (lower $Q/Q_{expected}$ means better fitting), to support the determination of factor number. The result, as shown in the new Figure S7, shows that increasing factor number from 3 to 4 and from 4 to 5 lowered the $Q_{true}/Q_{expected}$ value in a much greater extent when comparing to increasing the factor number from 5 to 6, 6 to 7 and 7 to 8. This indicates the 5-factor solution is most suitable for source interpretation.

Revisions is made as below:

Line 106 – 114: In this work, vehicular contributions to PM$_{2.5}$ are quantified by PMF analysis using the EPA PMF 5.0 software (Norris et al., 2014). PMF is a receptor model that solves the chemical mass balance of a speciated sample data matrix by decomposing it into factor profiles and factor contributions with non-negative constraints, with the objective to minimize the objective function Q (Paatero and Tapper, 1994; Paatero, 1997). The Q value represents the uncertainty weighted deviation between observed and modeled species concentrations.

Hourly concentrations of OC, EC, NO$_x$, CO and 12 selected VOC species from the entire monitoring period are considered in the PMF model for a single analysis. The VOCs, which were consistently detected above detection limit (> 80 % in each calendar year), include ethene, ethane, propane, propene, *i*-butane, *n*-butane, *i*-pentane, *n*-pentane, benzene, toluene, ethylbenzene and *m*-&*p*-xylene.

Line 232–236: Among various PMF solutions, the five-factor solution is the most interpretable for source identification and quantification. The drop in $Q_{true}/Q_{expected}$ value, which reflects the improvement in modeled species concentrations against measurements, is more significant when the factor number is increased by one from three to five compared to from five to eight (Fig. S7). This implies five factors are sufficient and suitable to explain the variations of input species data.

[Figure]

**Figure S7.** PMF performance in terms of the fitting between modeled and measured species concentrations expressed in $Q_{true}/Q_{expected}$ values, considering different factor numbers. Columns show the change in $Q_{true}/Q_{expected}$ values as the factor number increases by one (left axis). Markers show the $Q_{true}/Q_{expected}$ values in individual runs (right axis). The final solution of 5-factor run is indicated by an arrow.

Line 160, section 2.4: the author defined that the vehicle PM is the sum of total ambient EC and vehicle OC. Was there contribution of any solid-fuel burning (and cooking to OC) to EC and OC? If yes, how did the authors isolate these fractions using this vehicle OC/EC method, and assuming that total ambient EC was only from vehicle emissions? In addition, the authors simply considered the minimum OC/EC ratio as the vehicle OC/EC method. This part should be more clearly described, for example what this ratio is, how many selected samples associated with such minimum ratio, did the authors filter the potential influence of biomass burning and/or cooking (if applicable), etc.? These points should be discussed here.

**Response:** Study on source apportionment of EC at our roadside monitoring site has not been reported in the past. Certain amount of EC at this site could originate from nonvehicular sources such as biomass burning and coal combustion, but their contributions should be limited given the observed seasonal and diurnal patterns. Our PMF analysis shows that local vehicles contribute to 80–100% of EC in most sampling months, which further support this notion (Fig. S12b). Biomass burning and coal combustion are rare in our city area and their contributions are typically associated with regional transport in cold seasons. To isolate EC contributions from nonvehicular sources, measurement of additional chemical tracers is needed, which was not available in the study period. Still, given the strong evidence that EC at our roadside site is dominated by local traffic sources (e.g. based on diurnal variation patterns and PMF result), we believe the assumption that ambient EC is equivalent to vehicular EC is reasonable.

Details on the determination of $[OC/EC]_{vehicle}$ ratio and evidence to support its reliability have been given in section 3.2. Briefly, the minimum OC/EC ratios are determined on a monthly basis considering summer month data. The ratios range from 0.30 to 0.47, with $R^2$ between 0.56–0.96 (sample size $n = 18$–33). These ratios mostly occur during morning rush hour, when contribution from nearby cooking emissions is minimal.

We also make revision as below to clarify the use of the minimum OC/EC method:

Line 145–149: The [OC/EC]$_{vehicle}$ is determined using the minimum OC/EC ratio approach, in which the ambient OC in a certain lowest percentage range by OC/EC ratio are regressed on ambient EC, and the slope obtained represents the target ratio (Lim and Turpin, 2002). This minimum ambient OC/EC ratio is perceived to be of minimal contributions from secondary formation and nonvehicular primary sources, which typically have higher OC/EC ratio than VE (e.g. cooking emissions and biomass burning).

Line 180, section 3.1: For trend analysis/discussion in the manuscript, they could be further performed with statistical approach, for instance, the Mann-Kendall trend test. By using that, the magnitude of change rate for the trend can be quantified, along with the significance levels. Overall, it seems there were increasing trends during the beginning years, while the decreasing trends have been observed since around 2013? In addition, it seems that there was a rapid decrease in those trends during around 2013-2015 (?), while a slowdown decrease in those trends were observed during recent years. Therefore, those points might be further discussed and explain possible reasons.

**Response:** We thank the reviewer for the suggestion of using statistical approach to examine the trends reported in this work. The Mann-Kendall test can indicate whether a monotonic increasing/decreasing trend exist at certain confidence interval, while the Sen's slope can subsequently be used to estimate the rate of change of the monotonic trend if it exists. However, the OC/EC time series presented in this study appear to consist of both upward and downward trends in different periods where the transition point is difficult to locate, thus isolation of a specific period for statistical analysis would be arbitrary, so does the statistical results. We therefore attempt to only describe the distinctive features of the trends without further performing the statistical analysis.

We believe what the reviewer referring to is the trends for EC and NO$_x$ shown in Fig. 1. We agree that there could be a slight increase in concentration from 2011 to 2013, followed by a rapid drop by the end of 2014. While we are unclear about the reason behind the change in the beginning years, the sudden drop near the end of 2014 (especially for NO$_x$) could be attributed to road blockage of several major roads near MK AQMS for ~2 months due to a major protest. It is also surprising to notice the EC level rebound in the last two months. Unfortunately, the monitoring has ceased since that period, and we are unsure whether that reversing trend continued or not. Twenty four-hour filter-based speciation data from this site as part of Hong Kong's PM$_{2.5}$ network in recent years (have not been released to the public yet) will help fill in this gap.

Lines 186 – 190: I think air mass back-trajectory analysis could be applied here to support those statements. By this method, the authors will be able to investigate the different concentration levels and/or sources of OC and EC associated with different air mass origins/clusters.

**Response:** Hong Kong is located in the coastal area facing the South China Sea to the south and mainland China to the north. This geographical feature gives Hong Kong a typical monsoon climate that results in contrasting prevailing wind directions in different seasons, and therefore seasonal characteristics in level and composition of PM$_{2.5}$, including elevated organic mass during winter. This feature has been validated by air mass backward trajectory analysis and documented in a number of past studies (e.g. Louie et al., 2005; Hagler et al., 2006; Huang et al., 2014). We therefore do not repeat the analysis in this work. Instead, we provide the references to allow interested readers to look for further details.

Revision is made in the main text as below:

Line 160–162: During the fall/winter monsoon season, the prevailing northeasterly wind transports pollutants from the continental area to HK, while in summer the prevailing southerly wind carries clean air mass from the sea (Louie et al., 2005; Hagler et al., 2006; Huang et al., 2014).

Lines 190 – 192: Did the authors have any evidence to prove these reasons?

**Response:** We thank the reviewer for raising this question. We do observe the seasonal variation in mixing height in Hong Kong as shown in Figure R1 below. However, as we re-examine the OC/EC data, the EC does not exhibit discernible seasonal variation over the years. This lack of seasonality is also reported in Louie et al. (2005). These pieces of evidence suggest the variation in mixing height is indeed unlikely to produce sufficiently large impact on the OC concentration in our study area.

[Figure]

**Figure R1.** Trend in mixing height in Hong Kong during 11:00 (local time) recorded by the Global Telecommunication System of the World Meteorological Organization. Each data point represents the monthly average. Shaded area represents one standard deviation for the hourly data.

As for the effect of partitioning of SVOC, while we do not have direct measurement-based evidence for support, Lee et al.'s (2017) work does show that submicron vehicle-related organic aerosol (VE-OA) in the same study location tends to partition in the gas phase during summer, contributing to a 40% reduction in VE-OA mass compared to spring (2.0 vs. 3.5 µg m$^{-3}$). This implies gas-particle partitioning of SVOC could be an important factor contributing to the seasonal variation in OC concentration.

Revision is made in the main text as below:

Line 162–167: Another plausible reason for the elevated OC observed in wintertime is the enhanced partitioning of semivolatile organic compounds (SVOCs) into particle phase due to lower temperature and higher organic aerosol loading. Previous study at the same monitoring site shows that VE-related organic aerosol (derived from PMF analysis of organic aerosol mass spectra) decreases by 40 % in summer relative to spring despite consistency in traffic flow volume, pointing to a sizable influence of gas-particle partitioning of SVOCs (Lee et al., 2017).

Lines 192 – 193: It could help to further support the OC trend driven by wintertime OC when you separately show the six-year trends of monthly data for winter (DJF), spring (MAM), summer (JJA), and fall (SON). Did the authors find seasonal characterization for those OC and EC trends over the six-year period? Those new plots can be presented in supplement.

**Response:** We thank the reviewer for the suggestion. The trends during different seasons are added as the new Figure S3, which is also shown below. Instead of separating the seasons according to the reviewer's suggestion (i.e., DJF, MAM, etc.), we defined the seasons based on the subtropical climate feature of our location, which has longer summer and winter and shorter, transitional spring and fall (Louie et al., 2005). The new plot more clearly shows that the magnitude of OC reduction from 2011 to 2017 was the highest in winter.

Revision is made in the main text as below:

Line 168–169: It is noted that the winter OC had larger improvement than summer OC over the monitoring period, as shown in the season-specific trend plot in Fig. S3.

[Figure]

**Figure S3.** Inter-annual trends of (a) OC and (b) EC at MK AQMS from 2011 to 2017 during spring (mid-March to mid-May), summer (mid-May to mid-September), fall (mid-September to mid-November) and winter (mid-November to mid-March of next year). The measurement did not cover spring 2016 and 2017 and fall 2017, thus their results are not shown. Square marker and horizontal line within the box represent mean and median, respectively. Lower and upper bound of the box represent 25th and 75th percentile. Whiskers represent 5th and 95th percentile.

Lines 202-203: Based on these discussions above, it seems not fully convinced to conclude the less regional source influence on EC loadings rather than local traffic emissions. I guess, the similar EC diurnal cycles between work days and holidays/Sundays might reflect similar rush hours between the two types of days during a week. This could not sufficiently prove that the EC was more coming from local emissions. The different concentration levels of EC between the two types of days were also observed, however they weren't discussed. These similar diurnal patterns, along with different concentration levels, would be due to reduction in the total amount of traffic emissions over local and/or small-regional scales, however rush hours were overall not changed. In addition, as the NOx data was available in this work, it would be interesting to show correlations of EC versus NOx during work days and holidays, respectively. As commented above, air mass back trajectory analysis could also help to understand if there would have significant influence of regional sources on EC observed at the receptor site. Therefore, further discussions to support your statements should be extended.

**Response:** We thank the reviewer for the careful examination of the arguments. The main reason we think regional sources were not important to EC is that EC did not show seasonal variation as OC did. This implies regional air mass, which transports to our study area more frequently during winter, is enriched in OC to a much greater extent than EC, and thus the regional contribution to EC would be low. We add some description on the diurnal and workday–holiday patterns of EC observed during our long-term monitoring period to support VE as the major sources of EC. The revisions are as below:

Line 176–177: The absence of seasonal variation indicates local emissions dominated EC at this roadside site, and the impact of regional sources on EC, as opposed to OC, was limited.

Line 179–183: Such correlations persisted over the years, as shown in Fig. S4. Specifically, during workdays, EC concentration increased 4-fold from its minimum during small hours to ~ 4–8 µg C m$^{-3}$

during daytime. The corresponding increase was 2-fold for holiday period, consistent with the reduced traffic flow volume. These multiple lines of evidence indicate that EC at the site was mainly affected by local VE sources and less impacted by regional sources.

As suggested by the reviewer, we make scatter plot comparison between $NO_x$ and EC separately for workdays and holidays data, as shown below in Fig. R2. The two species had moderate correlation in both workdays and holidays. But since this correlation has been indirectly reflected by the similarity in their six-year trends shown in Fig. 1 in the main text, we will leave these plots in this response document for future reference only.

[Figure]

**Figure R2.** Scatter plot comparison between $NO_x$ and EC during (a) workdays and (b) holidays.

As mentioned in the previous response, the EC contributions from regional sources could be evaluated through comparing the EC concentrations in summer and winter given these two seasons have contrasting air mass origins associated with the prevailing wind direction. Again, the lack of seasonality in EC concentration supports that regional sources have a limited impact on EC concentration at our site. In addition, as our roadside site is surrounded by tall buildings, the wind information recorded at this micro-environment is subjected to uncertainty resulting from the heterogeneity in land surface, which would create bias in air mass backward trajectory analysis.

Line 241: It's not easy to justify the seasonality only based on time series of monthly data. It would be better if the authors could show monthly cycles and/or perform a seasonality significance test.

**Response:** The monthly cycles of *n*-butane concentration during 2011–2012 and 2014–2017 are shown in Fig. S5. Data from 2013 is not included because a major catalytic converter replacement program for LPG-fueled vehicles was undergoing and *n*-butane showed a precipitous drop during that year. From the figure, it is observed that the concentration remained almost the same throughout all months, supporting the seasonality is minimal. Revision is made in the main text as below:

Line 205–207: As shown in Fig. 1e, the *n*-butane level did not show obvious monthly variation over the years, supporting this species was predominantly emitted by local LPG vehicles (box-plot statistics of the monthly concentration are shown in Fig. S5).

[Figure]

**Figure S5.** Monthly variation of *n*-butane concentration during (a) 2011–2012 and (b) 2014–2017 at MK AQMS. Data from 2013 is not included because a major catalytic converter replacement program for LPG-fueled vehicles was undergoing and *n*-butane showed a precipitous drop during that year. Square marker and horizontal line within the box represent mean and median, respectively. Lower and upper bound of the box represent 25th and 75th percentile. Whiskers represent 5th and 95th percentile.

Lines 257-259: It would be also good to show diurnal variations of the OC/EC ratio to support the lowest ratios associated with the rush hours. As shown in Fig. S3, EC presents high concentration starting from around 7 AM – 6 PM. Could this suggest rush hours for EC spanning this time period? It might be also useful to check and discuss diurnal variations of OC concentrations, NOx and OC/EC ratios.

**Response:** We plot the study-wide diurnal variation of OC/EC ratio by season, as shown below in Fig. R3. It can be seen that the ratio is the lowest during the 7:00–10:00 am period. Indeed, the lowest 5 % OC/EC ratio data are mostly derived from this morning period. Although EC presents high concentration during 7 am–6 pm, the lowest OC/EC ratio mainly occurs during the morning rush hours because cooking emissions, which represent an important OC source at our site, are insignificant during this period. During mealtime in the afternoon and evening, OC/EC ratio increases sharply despite the consistently high EC level because of the OC contribution from cooking emissions. The OC/EC ratio from these time segments are therefore not suitable to represent VE.

The diurnal variation of OC has been given in Fig. S10 for the purpose of evaluating the PMF result, while that for NOx is given below as Fig. R4. These diurnal patterns (including OC/EC ratio as well) are very similar to those reported in our previous publication (Huang et al., 2014), which characterizes the impact of VE at the same roadside location for the first time. Given the focus of this extended work is on long-term trend analysis and separating VE contributions for different vehicle types, we tend to leave out the discussion on diurnal variation of various species. We will refer interested readers to our previous publication for detailed analysis and discussion of the diurnal variation patterns.

[Figure]

**Figure R3.** Diurnal variation of OC/EC ratio at MK AQMS during spring, summer, fall and winter. Square marker and horizontal line within the box represent mean and median, respectively. Lower and upper bound of the box represent 25th and 75th percentile. Whiskers represent 5th and 95th percentile.

[Figure]

**Figure R4.** Diurnal variation of NO$_x$ at MK AQMS during spring, summer, fall and winter. Square marker and horizontal line within the box represent mean and median, respectively. Lower and upper bound of the box represent 25th and 75th percentile. Whiskers represent 5th and 95th percentile.

Line 278: As commented above, did the authors perform only a single PMF run? Did you have any other PMF run tests, e.g., using seasonal runs? Based on these runs, did the authors have the same solution? And were the results from seasonal PMF runs consistent with the present results? Did the authors try to increase the number of PMF factors? How were those PMF solutions based these tests?

**Response:** The VE source contributions reported in this work is based on a single PMF run, which considers all data collected from the entire study period. We also performed the PMF analysis on subset of samples (by dividing the samples into three groups of equal sample size). The results from these additional PMF runs are consistent with the base PMF result, as discussed in sect. 3.3.2 (Fig. S9 and Fig. S11). We attempted to perform PMF analysis using winter and summer data separately. Five factors resembling those in the base PMF run are resolved in both seasons. However, the winter PMF did not apportion any OC and EC to the gasoline VE factor, which is not the case in the base PMF. This issue is not observed in summer PMF. A comparison in VE contributions between the base PMF and summer PMF is summarized in Fig. R5. Overall, the contributions derived from both PMF runs show excellent correlation ($R^2$ = 0.98 as shown in Fig. R5a–5f). The diesel contributions derived from the two PMF runs are comparable, while the gasoline contributions are higher in the summer PMF. From the summer PMF, the diesel contribution dominates the PM$_{vehicle}$ contribution as shown in Fig. R5g, in agreement

with the base PMF result. The reason why OC and EC are absent in gasoline VE factor in winter PMF warrant further investigation, but based on our PMF analyses, certainly considering all available data in PMF would yield most reasonable results.

[Figure]

**Figure R5.** Comparison of VE source contributions derived from PMF considering all data and PMF considering summer data only. Figure (a)–(c) show the results for diesel OC, EC and PM, respectively. Figure (d)–(f) show the results for gasoline OC, EC and PM, respectively. Figure (g) shows the relative contributions between diesel and gasoline VE to $PM_{vehicle}$, derived from summer PMF.

As mentioned earlier, we tested the PMF performance using a range of factor number from 3 to 8. We found that further increasing the factor number beyond 5 would not give significant improvement in $Q_{true}/Q_{expected}$ value. The additional factors resolved are also ambiguous. Take the 8-factor solution as an example, the additional factors include an *m-/p*-xylene factor, a toluene factor, and an unidentified factor. The presence of these excessive and unidentified factors hinders our interpretation of source analysis.

**References**

Hagler, G. S. W., Bergin, M. H., Salmon, L. G., Yu, J. Z., Wan, E. C. H., Zheng, M., Zeng, L. M., Kiang, C. S., Zhang, Y. H., Lau, A. K. H., and Schauer, J. J.: Source areas and chemical composition of fine particulate matter in the Pearl River Delta region of China, Atmos. Environ., 40, 3802–3815, 2006.

Huang, X. H. H., Bian, Q. J., Louie, P. K. K., and Yu, J. Z.: Contribution of vehicular carbonaceous aerosols to $PM_{2.5}$ in a roadside environment in Hong Kong, Atmos. Chem. Phys., 14, 9279–9293, 2014.

Lee, B. P., Louie, P. K. K., Luk, C., and Chan, C. K.: Evaluation of traffic exhaust contributions to ambient carbonaceous submicron particulate matter in an urban roadside environment in Hong Kong, Atmos. Chem. Phys., 17, 15121–15135, 2017.

Louie, P. K. K., Watson, J. G., Chow, J. C., Chan, A., Sin, D. W. M., and Lau, A. K. H.: Seasonal characteristics and regional transport of $PM_{2.5}$ in Hong Kong, Atmos. Environ., 39, 1695–1710, 2005.

---

## Author Response (AR2)

Title: Tracking Separate Contributions of Diesel and Gasoline Vehicles to Roadside PM2.5 Through Online Monitoring of Volatile Organic Compounds and PM2.5 Organic and Elemental Carbon: A Six-Year Study in Hong Kong
Author(s): Yee Ka Wong et al.
MS No.: acp-2020-9

*Response to Review Comments*

We thank the editor for carefully going through our manuscript and the supporting material. We have incorporated all the changes suggested by the editor. The changes are marked in blue in a marked-up manuscript version, which is appended at the end of this response document.

**Editor's comments**

The authors have reasonably well addressed the comments of the two anonymous referees and they have modified their manuscript accordingly. However, several alterations are needed for the Main text and Supplement before the manuscript can be published in ACP.

For the Main text:
Line 3: Replace "Compounds," by "Compounds and".
Line 26: Replace "of evidence-based" by "of an evidence-based".
Line 35: Replace "poses challenge" by "poses a challenge".
Lines 40, 48, 148, 221, 256, 361 and 365: Replace "e.g." by "e.g.,".
Line 64: Replace "2 h resolution" by "2-h-resolution".
Line 66: Replace "suit of" by "suite of".
Line 69: Replace "Here we" by "Here, we".
Line 80: Replace "effectiveness" by "the effectiveness".
Line 87: Replace "LPG respectively made" by "LPG made".
Line 88: Replace "29 % of" by "29 %, respectively, of".
Line 90: Replace "sources of" by "source of".
Line 93: Replace "concentrations data" by "concentration data".
Line 98: Replace "species including" by "species, including".
Line 100: Replace "exclude hourly" by "excluded hourly".
Line 102: Replace "are also" by "were also".
Line 109: Replace "Q value" by "Q-value".
Line 119: Replace "for non-roadside" by "for the non-roadside".
Line 151: Replace "Wu (2018)" by "Wu and Yu (2018)".
Line 161: Replace "air mass" by "air masses".
Line 163: Replace "into particle" by "into the particle".
Line 181: Replace "for holiday" by "for the holiday".
Line 185: Replace "Similar" by "A similar".
Line 190: Replace "is shown" by "are shown".
Line 196: Replace "was shown in" by "was observed in the".
Line 197: Replace "aerosol mass" by "the aerosol mass" and replace "two-thirds of" by "two-thirds of the".
Line 201: Replace "indicates that" by "indicate that".
Line 218: Replace "exhibit obvious" by "exhibit an obvious".

Line 223: Replace "air mass" by "air masses".

Line 225: Replace "colder" by "the colder" and replace "into particle" by "into the particle".

Line 254: Replace "show similar" by "show a similar".

Line 262: Replace "show that" by "shows that".

Line 264: Replace "in the all" by "in all".

Line 271: Replace "using box" by "using a box".

Line 289: Replace "its contribution" by "their contribution" and replace "to diesel" by "to that of diesel".

Line 298: Replace "reflects diesel" by "reflects the diesel".

Line 335: Replace "bus is" by "buses is".

Line 338: Replace "the trucks (i.e., DCVs) number is a factor of 2–3 higher than" by "the number of trucks (i.e., DCVs) is a factor of 2–3 higher than that of".

Line 339: Replace "vehicle" by "vehicles" and replace "than trucks" by "than the count of trucks".

Line 340: Replace "vehicle" by "vehicles".

Line 343: Replace "vehicle" by "vehicles".

Line 351: Replace "produce obvious" by "produce an obvious".

Line 519: Replace "on left" by "on the left".

Line 536: Replace "Vertical" by "The vertical".

Line 537: Replace "Green-shaded" by "The green-shaded".

For the Supplement:

Page S-1, line 1: Replace "Tracking" by "Supplementary Material for: Tracking".

Page S-1, line 2: Replace "Compounds," by "Compounds and".

Page S-1, line 6 above heading "S2. PMF Modeling Details": Replace "MDL" by "The MDL".

Page S-1, line 1 below heading "S2. PMF Modeling Details": Replace "MDL" by "the MDL".

Page S-1, line 3 below heading "S2. PMF Modeling Details": Replace "to EPA" by "to the EPA".

Page S-1, line 2 above heading "S3. PMF Uncertainty Analysis Results": Replace "concentration below" by "concentrations below the".

Page S-1, line 2 below heading "S3. PMF Uncertainty Analysis Results": Replace "each species" by "the species".

Page S-1, last line: Replace "Q value" by "Q-values".

Page S-2, line 1: Replace "Q value" by "Q-value" and replace "Q values" by "Q-values".

Page S-2, line 3: Replace "obtained is" by "obtained are".

Page S-2, line 13: Replace "Fig. S9a-c" by "Fig. S11a-c".

Page S-3, line 4 from bottom: Replace "47 mm" by "47-mm".

Page S-3, line 3 from bottom: Replace "2 h" by "2-h".

Page S-3, last line: Replace "1 h" by "1-h".

Page S-4, top line of the Table: Replace "recovery" by "recovery (%)".

Page S-4, last line: Replace "Note: No" by "Note: N/A indicates that no".

Page S-5, footnote of Table S3: Replace "Note: The" by "Note: The asterisk indicates that the".

Page S-9, line 2 of caption of Figure S7: Replace "factor number" by "factor numbers".

Page S-9, line 4 of caption of Figure S7: Replace "of 5-factor" by "of the 5-factor".

Page S-11, line 3 of caption of Figure S11: Replace "were split" by "was split".

[revised manuscript text omitted]